# TactileVAD: Geometric Aliasing-Aware Dynamics for High-Resolution Tactile Control

**Miquel Oller**    **Dmitry Berenson**    **Nima Fazeli**
Department of Robotics, University of Michigan
Ann Arbor, MI 48109, United States
{oller, dmitryb, nfz}@umich.edu
https://www.mmintlab.com/tactile-vad

**Abstract:** Touch-based control is a promising approach to dexterous manipulation. However, existing tactile control methods often overlook tactile geometric aliasing which can compromise control performance and reliability. This type of aliasing occurs when different contact locations yield similar tactile signatures. To address this, we propose TactileVAD, a generative decoder-only linear latent dynamics formulation compatible with standard control methods that is capable of resolving geometric aliasing. We evaluate TactileVAD on two mechanically-distinct tactile sensors, SoftBubbles (pointcloud data) and Gelslim 3.0 (RGB data), showcasing its effectiveness in handling different sensing modalities. Additionally, we introduce the tactile cartpole, a novel benchmarking setup to evaluate the ability of a control method to respond to disturbances based on tactile input. Evaluations comparing TactileVAD to baselines suggest that our method is better able to achieve goal tactile configurations and hand poses.

**Keywords:** Manipulation, tactile control, high-resolution tactile sensors

## 1   Introduction

Tactile sensing is a key enabler for dexterous robotic manipulation, allowing robots to perceive the objects they interact with and control their behavior. The ability to sense and respond to external forces and changes in grasped object poses during manipulation is essential for a broad range of tasks including grasping, pushing, sliding, and tool-use. These tactile perception-action loops provide accuracy and robustness in the face of uncertainty while complementing vision by providing feedback despite occlusions from the robot or environment.

In recent years, there has been a growing interest in developing high-resolution, vision-based, and collocated tactile sensors for robotic manipulation [1, 2, 3, 4, 5]. These sensors convert tactile signatures to visual formats (e.g., RGB or point clouds) and have enabled a variety of perception tasks including in-hand pose estimation [6, 7, 8], object identification [9, 10, 11, 12], and slip detection [13, 14]. However, these

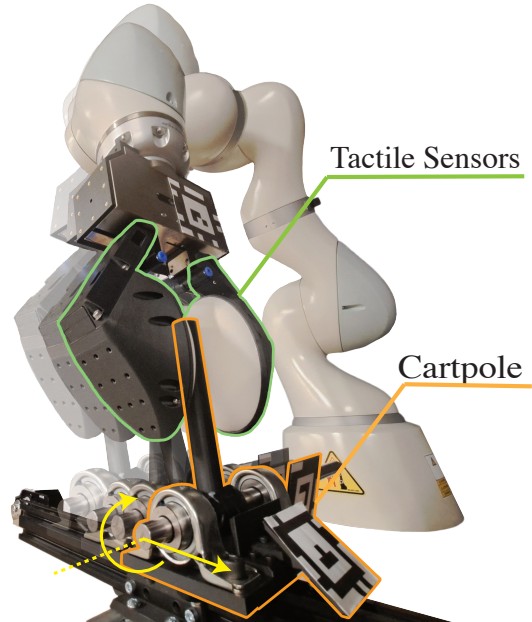

Figure 1: **Tactile Cartpole:** Our proposed benchmark task evaluates the robustness of tool grasping control subject to disturbances applied by the cart.

high-resolution tactile sensors, by virtue of their mechanical structure, introduce complex dynamics between the robot, grasped object, and the environment that must be accounted for in the control

7th Conference on Robot Learning (CoRL 2023), Atlanta, USA.

of the robot [15]. Current state-of-the-art tactile control methods rely on black-box modeling of the sensor-object-environment interaction coupled with sample-based control [16, 17, 18] or reinforcement learning [19, 20].

While these approaches have shown promise in leveraging tactile feedback for control, they often overlook a critical factor: tactile geometric aliasing. This aliasing occurs when different configurations or contact geometries produce indistinguishable tactile signals, leading to ambiguity in interpreting the underlying interactions. Neglecting this effect can hinder planning and control performance, as we show in this paper. To address these limitations, we introduce a tactile representation and controller that is able to disambiguate geometric aliasing. Our contributions are:

- TactileVAD, a generative decoder-only linear latent dynamics formulation for tactile control with high-resolution sensors that is robust to geometric aliasing,
- Demonstration of this method for two mechanically-distinct tactile sensors with different sensing modalities: SoftBubbles [1] (point cloud data) and Gelslim 3.0 [2] (RGB data),
- Tactile cartpole, a novel benchmarking setup for evaluating the stability of tool grasping control in the presence of external perturbations.

## 2 Related Work

**Tactile Control** In recent years, there has been a growing interest in the application of tactile sensors for various robotic manipulation tasks, including insertion [21], cloth manipulation [22], cable manipulation She et al. [18], or tool manipulation [23, 15]. Tactile sensors provide valuable information for controlling the interaction between the robot and a grasped object. Some existing methods focus on learning the dynamics of sensor-object interactions [16, 3] to utilize them for model-based control. However, the high dimensionality of the tactile signals poses a challenge in modeling these dynamics and using them for controls. To address this, a common approach is to project them into a lower-dimensional space [15, 19]. Despite the potential shown by these approaches, they frequently neglect the presence of ambiguity in tactile signals arising from geometric aliasing. In our work, we address this issue by constructing latent spaces capable of disambiguating aliased observations.

**Latent-Space Dynamics** The identification of system dynamics from raw observations plays a crucial role in the development of effective control strategies. Brunton et al. [24] proposed to identify the underlying dynamics of a non-linear system through sparse regression. Since observations are often high-dimensional, it is common to represent the system dynamics in a lower-dimensional space [25]. Many of these approaches employ encoder-decoder architectures, and one notable framework is Embed-to-Control (E2C) [26]. E2C formulates the latent as a Variational Auto-encoder with locally-linear latent space dynamics over conditional distributions. This work inspired subsequent research efforts. Jaques et al. [27] explored a more constrained parametrization of the latent dynamics, resulting in improved learning of the latent space for PD control laws. Furthermore, Okumura et al. [28] extended this approach to incorporate tactile sensing by learning a latent space projection for both tactile and visual inputs. Despite the success of encoder-decoder architectures, they can introduce aliasing in the latent space when faced with aliased observations. To overcome this limitation, our approach employs a decoder-only architecture and updates the latent space through back-propagation, effectively mitigating the adverse effects of aliasing.

**Decoder-Only Architectures** Decoder-only models have gained attention as an alternative to traditional encoder-decoder architectures. These models eliminate the need for an encoder by leveraging gradient-based optimization to infer the latent space representation directly from the observations [29]. Decoder-only architectures have exhibited promising results in learning structured latent spaces for high-dimensional data generation [30, 31, 32, 33]. A notable example of this is the Variational Auto-decoder (VAD) [34], which extends the well-known Variational Auto-encoder (VAE) [35] formulation to a decoder-only scenario. VAD incorporates a codebook that stores conditional latent distribution parameters, which are jointly optimized along with the decoder weights. This formulation regularizes and encourages a more structured and smooth latent space. In our work, we extend the VAD formulation by introducing dynamics through linear transitions in the latent space. Our proposed approach, TactileVAD, enables the generation of temporal trajectories of states.

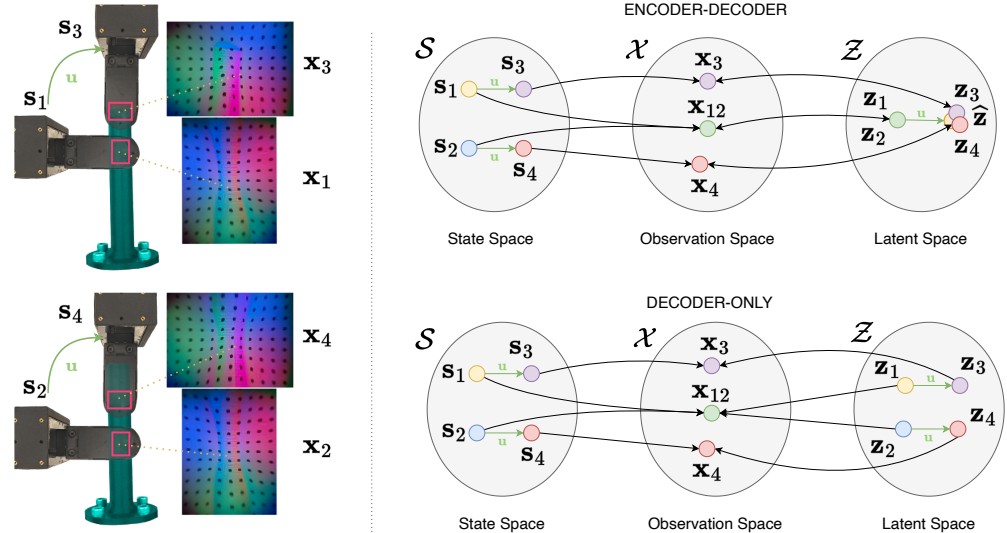

Figure 2: **Geometric Aliasing** (Left) illustrates the geometric aliasing problem of the tactile sensor reading while grasping an object with geometric symmetries. Observe that two different states $\mathbf{s}_1$ and $\mathbf{s}_2$ corresponding to different grasping locations result in the same sensor measurement $\mathbf{x}_1 = \mathbf{x}_2$. However, the same action $\mathbf{u}$ applied to both of them results in very different next-state observations. (Top Right) As a consequence of the aliasing in the observations, the mapping $\phi : \mathcal{X} \to \mathcal{Z}$ introduces ill-posed latent space. (Bottom Right) modelling only the $\psi : \mathcal{Z} \to \mathcal{X}$ results in better latent spaces.

## 3 Problem Formulation

In this paper, we consider the problem of controlling a system with unknown dynamics of the form $\mathbf{s}_{t+1} = f_{\text{dyn}}(\mathbf{s}_t, \mathbf{u}_t)$. Here, $\mathbf{s}_t \in \mathcal{S} \subset \mathbb{R}^{n_s}$ represents the system state at time $t$ consisting of both robot and environment configurations. The control actions $\mathbf{u}_t \in \mathcal{U} \subset \mathbb{R}^{n_a}$ correspond to the robot's end-effector motions. We assume that the true system state $\mathbf{s}_t$ is not directly measurable, but we have access to observations $\mathbf{x}_t = \text{obs}(\mathbf{s}_t) \in \mathcal{X} \subset \mathbb{R}^{n_x}$. Specifically, in our study, these observations $\mathbf{x}_t$ are obtained using vision-based tactile sensors and can take the form of point clouds or RGB images. We further assume that $\mathbf{x}$ may suffer from observation aliasing, where different system states $\mathbf{s}_1, \mathbf{s}_2 \in \mathbb{R}^{n_s}$, $\mathbf{s}_1 \neq \mathbf{s}_2$ can result in the same observations $\mathbf{x}_1 = \text{obs}(\mathbf{s}_1) = \mathbf{x}_2 = \text{obs}(\mathbf{s}_2)$. This aliasing can occur due to geometric symmetries and limited sensor measurement range. This situation can lead to an ill-posed inference problem, as the indistinguishable observations $\mathbf{x}_1$ and $\mathbf{x}_2$ result in ambiguity in determining the underlying dynamics. Fig. 2 (Left) illustrates this. The goal of this problem is to determine a sequence of actions $\mathbf{u}_1, \ldots, \mathbf{u}_T$ that can drive the system from some initial $\mathbf{s}_0$ to a desired system state $\mathbf{s}_g \in \mathcal{S}$ in the presence of observation aliasing.

## 4 Methods

### 4.1 Variational Auto-Decoder Linear Latent Dynamics

The key idea of our method is to learn a model of the underlying system dynamics in a lower-dimensional latent space $\mathcal{Z}$ such that: (i) observations $\mathbf{x}_t$ are reconstructed; (ii) the latent space dynamics are linear, smooth, and structured; and (iii) aliased states are identified and disambiguated.

One common approach is to learn a mapping function $\phi : \mathcal{X} \to \mathcal{Z}$ that projects high-dimensional observations $\mathbf{x} \in \mathbb{R}^{n_x}$ to a lower-dimensional latent space $\mathbf{z} \in \mathbb{R}^{n_z}$ where $n_z \ll n_x$. The dynamics in this latent space are globally-linear and given by $\mathbf{z}_{t+1} = f_{\text{lat\_dyn}}(\mathbf{z}_t, \mathbf{u}_t) = \mathbf{A}\mathbf{z}_t + \mathbf{B}\mathbf{u}_t$ where $\mathbf{A} \in \mathbb{R}^{n_z \times n_z}$, $\mathbf{B} \in \mathbb{R}^{n_z \times n_u}$. This approach typically employs an encoder-decoder architecture [26, 19, 25]. However, we observe that formulating the mapping from observation space to latent space $\phi : \mathcal{X} \to \mathcal{Z}$ introduces ill-defined latent spaces when aliased observations are present in the dynamics. With reference to Fig. 2 (Right), consider two distinct states $\mathbf{s}_1 \neq \mathbf{s}_2$ that result

in aliased observations $\mathbf{x}_1 = \text{obs}(\mathbf{s}_1) = \mathbf{x}_2 = \text{obs}(\mathbf{s}_2)$. Additionally, assume the existence of an action $\mathbf{u} \in \mathcal{U}$ that leads to non-aliased states: $\mathbf{s}_3 = f_{\text{dyn}}(\mathbf{s}_1, \mathbf{u})$ and $\mathbf{s}_4 = f_{\text{dyn}}(\mathbf{s}_2, \mathbf{u})$ where $\mathbf{s}_3 \neq \mathbf{s}_4$. Let $\mathbf{x}_3 = \text{obs}(\mathbf{s}_3)$ and $\mathbf{x}_4 = \text{obs}(\mathbf{s}_4)$ be the corresponding observations, such that $\mathbf{x}_3 \neq \mathbf{x}_4$. The resultant latent representations are denoted as $\mathbf{z}_i = \phi(\mathbf{x}_i)$. In such cases, the mapping $\phi$ will force $\mathbf{z}_1 = \mathbf{z}_2$ since $\mathbf{x}_1 = \mathbf{x}_2$, and therefore they are compelled to produce the same resultant dynamics $\hat{\mathbf{z}}$: $f_{\text{lat\_dyn}}(\mathbf{z}_1, \mathbf{u}) = f_{\text{lat\_dyn}}(\mathbf{z}_2, \mathbf{u}) = \hat{\mathbf{z}}$. However, the dynamics should yield distinct outcomes: $f_{\text{lat\_dyn}}(\mathbf{z}_1, \mathbf{u}) = \mathbf{z}_3 \neq \mathbf{z}_4 = f_{\text{lat\_dyn}}(\mathbf{x}_2, \mathbf{u})$. This situation leads to an ill-posed problem, as the indistinguishable observations $\mathbf{x}_1$ and $\mathbf{x}_2$ result in ambiguity in determining the underlying dynamics. Figure 2 (Top Right) visualizes this effect.

To address this, we propose an alternative approach that focuses exclusively on modeling the mapping from the latent space to observation space, denoted as $\psi : \mathcal{Z} \rightarrow \mathcal{X}$. By adopting this surjective mapping $\psi$, we can have multiple latent vectors $\mathbf{z}$ mapped to the same observation $\mathbf{x}$. Each of these $\mathbf{z}$ will evolve differently when action $\mathbf{u}$ is taken. Once the corresponding outcomes are perceived, we can use an inference via optimization procedure (Sec. 4.3) to disentangle the aliased observations. Fig. 2 (Bottom Right) provides an illustration of this concept. Our objective is to jointly learn the mapping from latent space to observation space, $\psi$, and the latent space dynamics $f_{\text{lat\_dyn}}$ from state observation transitions $(\mathbf{x}_1, \mathbf{u}_1, \mathbf{x}_2, \ldots, \mathbf{x}_{T-1}, \mathbf{u}_{T-1}, \mathbf{x}_T)$. To this end, we employ a decoder-only architecture. Additionally, we choose to model the latent space as conditional distributions, which leads to more structured and smooth learned latent spaces. Consequently, our model takes the form of a VAD with globally-linear latent dynamics, in contrast to E2C, which is formulated as a VAE. The latent space dynamics are of the form:

$$\mathbf{z}_t \sim Q(Z_t) = \mathcal{N}(\boldsymbol{\mu}_t, \boldsymbol{\Sigma}_t); \quad \mathbf{z}_{t+1} \sim Q(Z_{t+1}|Z_t, \mathbf{u}_t) = \mathcal{N}(\mathbf{A}\boldsymbol{\mu}_t + \mathbf{B}\mathbf{u}_t, \mathbf{A}\boldsymbol{\Sigma}_t\mathbf{A}^\top)$$

We have chosen this parametrization of latent dynamics to ensure compatibility with standard control methods like LQR. Section 4.4 describes our control formulation.

In our method, we eliminate the need for an encoder by maintaining a codebook of size $C$ composed of normal distribution parameters $\{\boldsymbol{\mu}_i, \boldsymbol{\Sigma}_i\}_{i=1\ldots,C}$. The size of the codebook is given by the number of states in the training data, since for each training state, we initialize a codebook element. These parameters are optimized jointly with the decoder's, and the dynamics parameters via back-propagation. The training process is described in detail in Section 4.2. During the inference, when mapping from observation space to latent space, we employ an optimization-based approach, as explained in 4.3. This inference procedure enables us to accurately determine the latent space representation based on the observed data. Furthermore, it allows us to generate multiple likely candidates for an aliased observation, providing a more comprehensive understanding of the underlying latent dynamics.

## 4.2 TactileVAD Training

We train our model from a dataset of $N$ trajectories of $T$ state-action transitions of the form $\mathcal{D} = \{(\mathbf{x}_1, \mathbf{u}_1, \mathbf{x}_2, \ldots, \mathbf{x}_T, \mathbf{u}_T, \mathbf{x}_{T+1})\}_{n=1}^N$. Our goal is to jointly optimize the latent space parameters

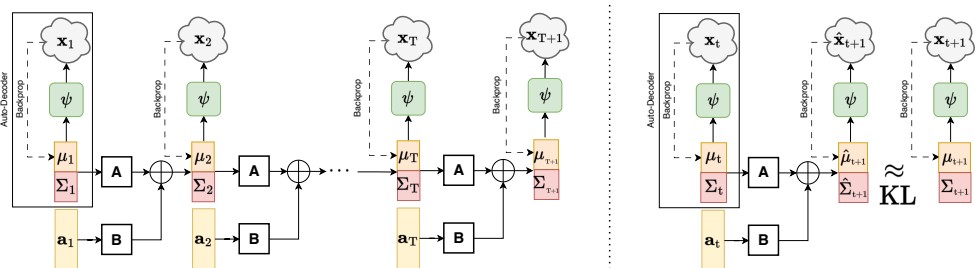

Figure 3: **VAD Latent Linear Dynamics Model** (Left) TactileVAD for trajectory generation of length $T$ via latent space linear dynamics propagation. (Right) single-step training with reconstruction, prediction, and latent space consistency.

$\{\boldsymbol{\mu}_i, \boldsymbol{\Sigma}_i\}_{i=1\ldots,(T+1)N}$, the transition dynamics $\mathbf{A}$ and $\mathbf{B}$, as well as the parameters $\theta$ that define the mapping from latent space to observation space $f_{\text{dec}}$. The training procedure is outlined in Algorithm 1. To jointly learn this, we use a loss function composed of four weighted terms:

$$\mathcal{L} = \alpha_{\text{rec}}\mathcal{L}_{\text{rec}}(\mathbf{x}_t, q_t) + \alpha_{\text{pred}}\mathcal{L}_{\text{pred}}(\mathbf{x}_{t+1}, \hat{q}_{t+1}) + \alpha_{\text{cons}}\mathcal{L}_{\text{cons}}(\hat{q}_{t+1}, q_{t+1}) + \alpha_{\text{reg}}\mathcal{L}_{\text{reg}}(q_t, q_{t+1})$$

where (i) $\mathcal{L}_{\text{rec}}$ imposes that we are able to reconstruct the current observation from the current latent distribution, (ii) $\mathcal{L}_{\text{pred}}$ imposes that we are able to generate the next observation from the predicted distribution dynamics given by the control action $\mathbf{u}_t$, (iii) $\mathcal{L}_{\text{cons}}$ imposes that the predicted latent distribution matches the distribution, and (iv) $\mathcal{L}_{\text{reg}}$ imposes a prior distribution on the latent space which regularizes the latent space. See Appendix A.1 for more details about the training and the loss formulation.

## 4.3 Embedding via Inference Search

Decoder-only architectures exclusively model the mapping from latent space to observation space $\psi : \mathcal{Z} \rightarrow \mathcal{X}$. Unlike encoder-decoder architectures, which can infer the latent representation $\mathbf{z}_t$ by forward-passing the encoder, decoder-only architectures do not directly model the mapping $\phi : \mathcal{X} \rightarrow \mathcal{Z}$. Instead, they employ an inference-based encoding approach to obtain a set of latent vectors corresponding to a given observation. Here, we infer $\mathbf{z}$ values by fixing the decoder weights $\theta$ and performing gradient descent on the training loss to compute the most likely $\mathbf{z}$ values that produced $\mathbf{x}$, similar to the procedure proposed in [30].

$$\hat{\mathbf{z}} = \arg\min_{\mathbf{z}} \mathcal{L}_{\text{state}}(f_\theta^{\text{dec}}(\mathbf{z}), \mathbf{x})$$

Although this inference approach is more computationally expensive than a forward encoding pass, it offers two main advantages: (i) better generalization to unseen data and distribution shifts and (ii) generation of multiple candidate latent states that result in the same observation. The ability to generate multiple candidates helps disambiguate aliased observations by capturing the inherent uncertainty in the latent space representation. To generate multiple candidates, we initialize a set of values for the latent elements to be optimized. In this work, we combine random sampling with priors. Specifically, given the first tactile observation, we initialize the candidates by uniformly sampling the latent space and the top $n$ elements in our codebook that are most similar to the current observed state. We add Gaussian noise to the codebook values to obtain a diverse set of initial candidates. During optimization, these latent elements will converge to various local minima, resulting in a diverse set of latent candidates. To maintain temporal consistency over the trajectory, we propagate candidates using the latent space dynamics with additive Gaussian noise. Generating multiple candidates is easily parallelized, resulting in only a minor addition to computation (see Sec B.1.1).

## 4.4 Low-level Control with VAD Dynamics Model

The goal of the controller is to drive the system state $\mathbf{s}_t$ to a desired target state $\mathbf{s}_g$. Given that we only have access to an aliased observations $\mathbf{x}_t$ of the system state $\mathbf{s}_t$, we formulate our controller in the learned latent space $\mathcal{Z}$ since it does not suffer from aliasing and has the added convenience of linear transition dynamics. Given a quadratic cost, we formulate a discrete time LQR controller:

$$
\begin{aligned}
\text{(P1)} \min_{\mathbf{u}} \quad & \sum_{t=1}^{\infty} \bar{\mathbf{z}}_t^\top \mathbf{Q}\bar{\mathbf{z}}_t + \mathbf{u}_t^\top \mathbf{R}\mathbf{u}_t \\
\text{s.t.} \quad & \bar{\mathbf{z}}_t = \mathbf{z}_t - \mathbf{z}_g \qquad t = 1, \ldots, \infty, \\
& \mathbf{z}_{t+1} = \mathbf{A}\mathbf{z}_t + \mathbf{B}\mathbf{u}_t \quad t = 1, \ldots, \infty
\end{aligned}
$$

with $\mathbf{Q} \succeq 0$, $\mathbf{R} \succ 0$. The optimal controller is computed by solving the discrete-time algebraic Riccati equation (DARE), yielding $\mathbf{u}_t = -\mathbf{K}\bar{\mathbf{z}}$ [36]. Here, we choose $n_z = n_u$. When $n_z > n_u$, the resulting system dynamics are underactuated and the controller may not have a stable solution. Algorithm 2 summarizes our controller formulation.

One consideration is that our approach can generate multiple latent candidates for a given observed state. To integrate this capability with controls, we maintain a history of candidates, initialized as

described in section 4.3. At each step, we propagate the dynamics on the candidate set and evaluate the similarity of their decoded state observations with the actual observed sequence. We select the candidate with the lowest state trajectory loss against the observed state observation trajectory. This gives the current latent value $\mathbf{z}_t$ that the controller takes to produce the optimal action.

## 5 Experiments and Results

We demonstrate the effectiveness of our proposed approach to tactile control in 4 tasks: (i) an aliased-rich simulation task with discrete states, where many of them exhibit aliasing, (ii) a real-world tactile control for static tool grasping with the Soft Bubbles tactile sensors and (iii) Gelslims 3.0 sensors, and (iv) the tactile cartpole: a novel quasi-dynamic tactile control bench-marking task for tool grasping under external perturbations inspired by the classic cartpole control task.

### 5.1 Baselines

We benchmark our method with 3 baselines, including 2 encoder-decoder approaches and one decoder-only approach. All methods have linear latent dynamics. See Supp. B.1 for details.

**Auto-encoder Dynamics (AE)** This baseline is a common approach that uses auto-encoders to shape the latent space [25]. The encoder projects high-dimensional observations into the latent vector space where dynamics are propagated. Predictions are then projected back to the observation space using the decoder. Controls are formulated in the low-dimensional latent space.

**Embed to Control (E2C)** Embed to control [26] is a popular framework for latent space dynamics that extends the auto-encoder formulation to the variational setting. The resulting latent spaces are distributions over vector spaces and have been shown to perform well for control tasks. For consistent evaluation, we use the global-E2C formation (globally linear dynamics).

**Auto-decoder Dynamics (AD)** This decoder-only framework optimizes a latent codebook simultaneously with the dynamics parameters and the decoder parameters. The main difference from our proposed approach is in the latent space, which is not a distribution. Here, the optimized codebook is not distribution parameters, but the latent vectors directly.

### 5.2 Moving Block Control

This task aims to demonstrate the impact of aliasing when modeling the dynamics on an aliased-rich task with discrete states. Fig. 5 provides a visualization of this task. The objective is to control the planar position of the rectangular object (in yellow) from image observations, where the object undergoes 2D translation motions. The states $\mathbf{s}_t$ represent the object's position in the 2D plane, while

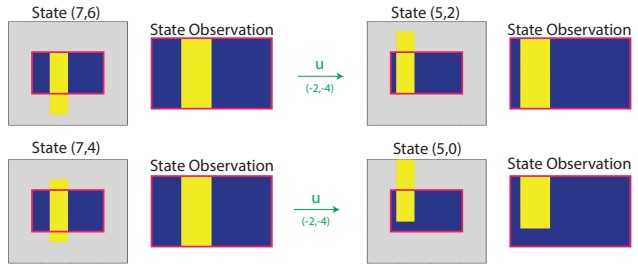

Figure 4: **Moving Block Setup** On the left, two different block locations result in the same observation. However, the same action applied to both states resulted in different next states (right).

the observations $\mathbf{x}_t$ capture only a partial view of the full state (depicted as the blue subpart around the center). Fig. 5 also shows an example of observation aliasing, where two different states lead to the same observation due to the small sensed area.

We benchmark our approach and compare it to the baselines. The goal is to control the block position to reach a desired target location. We evaluate the reconstruction and prediction quality as well as the LQR-controlled performance of the resulting dynamics models. Tab. 1 summarizes the results. We note that the decoder-only models (AD and VAD) result in much lower reconstruction error and prediction errors. This can be explained because the learned latent space from decoder-only models does not suffer from aliasing. In particular, our approach (VAD) consistently obtains the best scores in long-horizon predictions, which suggests that our method results in better-structured latent spaces (see Appendix C.1). Furthermore, our approach results in better controlled achieved pose (Tab. 1 first column). See Appendix B.2 for more details.

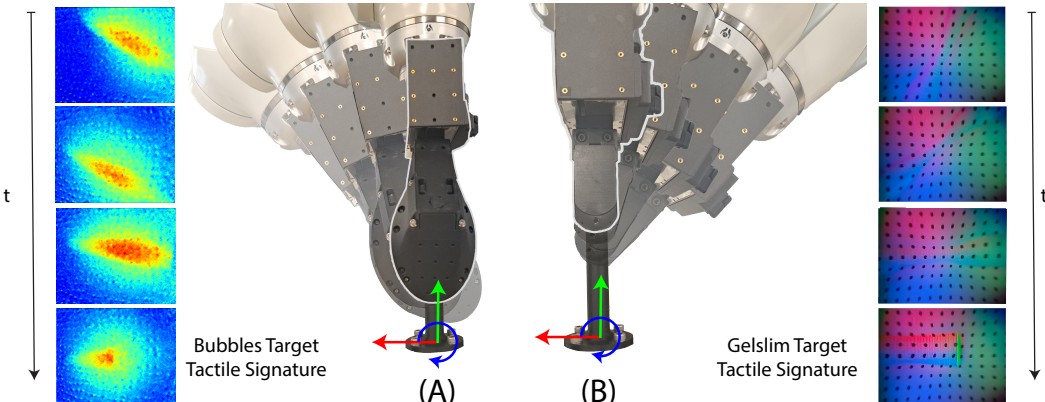

Figure 5: **Tactile Rod Grasping** (A) Rod grasping with Soft Bubbles where tactile signatures are encoded as depthmaps. (B) Rod grasping with Gelslims 3.0 where tactile signatures are encoded as RGB images.

## 5.3 Tactile Rod Grasping

The goal of this task is to control the relative pose between the robot end-effector and a grasped object from only tactile data. The object being grasped is a 3D-printed rod designed to simulate a tool handle. This rod is fixed in the scene to facilitate easy reset and data collection. It is important to note that the controller does not have access to

| Method | LQR Pose Errors [px] | | Reconstruction Error ↓ | Prediction Error ↓ ( $\cdot 10^{-3}$ ) | | |
|---|---|---|---|---|---|---|
| | Mean ↓ | Std ↓ | | 1 step | 5 step | 10 step |
| AE | 2.13 | 4.36 | $2.21 \cdot 10^{-3}$ | 21.19 | 318.4 | 1045.3 |
| E2C | 4.01 | 3.02 | $7.07 \cdot 10^{-2}$ | 102.4 | 110.6 | 107.7 |
| AD | 2.90 | 3.22 | $\mathbf{2.41 \cdot 10^{-7}}$ | 2.231 | 40.57 | 78.77 |
| VAD (ours) | **0.53** | **2.37** | $1.14 \cdot 10^{-5}$ | **0.726** | **30.74** | **62.96** |

Table 1: **Moving Block Evaluation:** LQR Pose Errors = error between an unaliased target pose and the pose achieved after 10 control steps following LQR control law. Reconstruction Error and Prediction Error for 500 test data trajectories propagating latent dynamics.

the robot pose with respect to the grasped object; it solely relies on tactile measurements. The control actions are the changes in position and orientation of the robot end-effector. The control actions are constrained to a box, requiring the robot to perform multiple grasps to achieve a desired relative pose.

We evaluate our methods on two tactile sensors with distinct mechanical properties and sensing modalities [15]: (i) the Soft Bubbles are composed of two inflated membranes that deform when in contact. This sensor exhibits large deformations that are sensed via a depth camera. We also evaluate our methods on the Gelslims 3.0 (ii) which are more rigid than the soft-bubbles and the sensor deformations are sensed via RGB images.

To train the models, we collect 500 trajectories of robot-rod interactions under random actions for each sensor. The length of these trajectories is 10. The rod used in these interactions has a cylindrical dimension of 120mm in length and a diameter of 20mm. In the case of the Soft Bubbles, the observed states are encoded as deformation maps of the sensor, representing the difference between the current sensor depth map and a reference undeformed state. On the other hand, for the Gelslims sensor, the observations are grayscale image differences between the current state and a reference undeformed state.

To assess the performance of the learned methods, we execute 20 trajectories using an LQR controller, starting from random configurations of the robot grasping the rod. The objective is to achieve a specific robot-rod configuration, where the rod is grasped at the center of the sensor and aligned with the end-effector. This goal configuration is provided as a non-aliased tactile observation. The evaluation was conducted using three different rods: the same 20mm rod used during training, as well as two novel rods of sizes 15mm and 30mm, to assess the generalization capability of the methods to varying geometries. The pose error between the final achieved configuration and the target configuration was computed as a measure of performance. Detailed information on the metrics computation can be found in Appendix B.3. Tables 2 and 6 present a summary of these metrics.

| Method | Bubbles Rod Pose Errors [mm] | | | | | | Gelslim Rod Pose Errors [mm] | | | | | |
|---|---|---|---|---|---|---|---|---|---|---|---|---|
| | 20mm Rod (Train) | | 15mm Rod | | 30mm Rod | | 20mm Rod (Train) | | 15mm Rod | | 30mm Rod | |
| | Mean ↓ | Std ↓ | Mean ↓ | Std ↓ | Mean↓ | Std↓ | Mean ↓ | Std ↓ | Mean ↓ | Std ↓ | Mean↓ | Std↓ |
| AE | 41.79 | 37.69 | 27.27 | 36.92 | 53.86 | 52.44 | 16.77 | 22.67 | 27.27 | 36.92 | 11.66 | 16.29 |
| E2C | 25.29 | 31.65 | 77.11 | 58.14 | 31.08 | 23.82 | 14.45 | 15.15 | 18.22 | 10.26 | 16.01 | 9.69 |
| AD | 58.05 | 36.27 | 65.46 | 31.37 | 62.26 | 39.04 | 64.13 | 18.00 | 40.61 | 36.96 | 80.22 | 11.74 |
| VAD (ours) | **5.23** | **4.06** | **10.91** | **3.99** | **7.68** | **2.67** | **8.54** | **16.01** | **8.98** | **7.27** | **10.39** | **19.83** |

Table 2: **Tactile Rod Grasping Evaluation:** Measuring the error between the planar pose achieved for 20 control steps and the target (rod vertical). For each sensor, we test on the training rod (20mm diameter) as well 2 unseen rods (15mm and 30mm diameter). Statistics are reported over 20 trials per method and rod.

Notably, our method consistently yielded lower errors in achieving the desired tactile configuration and pose, demonstrating its effectiveness for both known and unknown objects, as well as across different sensor modalities.

## 5.4 Tactile Cartpole

The goal of this task is to demonstrate the utility of our method as a low-level controller for tool manipulation tasks. To this end, we combine our learned tactile low-level controller to correct the relative robot-tool position with a high-level controller that controls tool orientation.

The setup is composed of a cylindrical rod connected via a revolute joint to an actuated cart on a rail. The rod is grasped by the robot and simulates the tool handle. Fig. 1 illustrates this task. By moving the cart, we perturb the grasped tool pose, serving as an automated mechanism. The

| Method | Number of Steps | | Rod Angular Mean |
|---|---|---|---|
| | Mean↑ | Std ↓ | Error [rad] ↓ |
| AE | 11.84 | 7.42 | 0.2949 |
| E2C | 8.12 | 3.17 | 0.2682 |
| AD | 11.73 | 7.95 | 0.2057 |
| VAD (ours) | **73.06** | **23.25** | **0.09267** |

Table 3: **Cartpole Control Evaluation:** Reporting the number of perturbations steps without letting the tool slip out of hand or dropping the tool and the rod angle error with respect to the target vertical rod. Statistics are reported over 20 trials per method.

goal of the robot is to keep the rod vertical and aligned with its end-effector. Therefore, the cartpole motions force the robot to correct and adapt to the resulting perturbed tool pose.

We compare our method with the baselines, all sharing the same high-level controller, differing only on the low-level robot-rod align motion. We evaluate the time the robot is able to keep the rod vertical without dropping it as well as the average angle error, Tab. 3. Our method considerably outperforms the baselines and is able to maintain the tool vertical for longer periods of time and track the desired relative orientation significantly more successfully. This suggests that our approach is able to better generalize to novel tasks.

## 6 Discussion and Limitations

One limitation of decoder-only architectures is that obtaining the latent representation requires multiple passes through the decoder using inference-via-optimization, which can be slower compared to encoder-decoder architectures that only require a single forward pass (see Appendix B.1.1). Another limitation of our approach is its limited exposure to cylindrical rods. Since tools can have various handle geometries such as rectangular or curved handles, generalizing the learned model becomes a challenge. To address this, we propose learning separate models for each primitive handle geometry. At runtime, a handle geometry-aware classifier can be used to select the appropriate controller based on the current grasped tool geometry, enabling better adaptability to different handle shapes.

Finally, both the Gelslims and Soft Bubbles tactile sensors are smooth elastomers despite their different mechanical properties. In our approach, we employ decoder architectures based on convolutional neural networks (CNNs) to generate the sensor data. However, it is worth noting that these CNN-based decoders may not fully capture the inherent characteristics of the sensors, like smoothness and continuity. To overcome this limitation, alternative methods can be explored, which specifically incorporate smoothness constraints on the sensor geometry. By adopting such methods that better capture the three-dimensional geometry of the sensors, more accurate predictions can be achieved, thereby improving the overall performance and reliability of the system.

## Acknowledgments

This research project is supported by Toyota Research Institute under the University Research Program (URP) 2.0 and "la Caixa" Fellowship Program (ID 100010434).

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

# Appendix A   Implementation Details

## A.1   TactileVAD Implementation and Training Details

In this section, we describe how we train the TactileVAD. Algorithm 1 describes in detail the training procedure. Note that the algorithm is similar to the VAD training algorithm 3 but with latent space dynamics over trajectories. As a result, the TactileVAD training optimizes (i) the decoder parameters $\theta$, (ii) the latent dynamics parameters $\mathbf{A}, \mathbf{B}$, and (iii) the latent conditioned distribution parameters $\boldsymbol{\mu}$ and $\boldsymbol{\Sigma}$. The optimization cost function is composed of 4 terms, and it is described in detail in Sections 4.2 and B.1.

The loss function is composed of four weighted terms:

$$\mathcal{L} = \alpha_{\text{rec}}\mathcal{L}_{\text{rec}}(\mathbf{x}_t, q_t) + \alpha_{\text{pred}}\mathcal{L}_{\text{pred}}(\mathbf{x}_{t+1}, \hat{q}_{t+1}) + \alpha_{\text{cons}}\mathcal{L}_{\text{cons}}(\hat{q}_{t+1}, q_{t+1}) + \alpha_{\text{reg}}\mathcal{L}_{\text{reg}}(q_t, q_{t+1})$$

Each of the losses is formulated as follows:

$$\mathcal{L}_{\text{rec}}(\mathbf{x}_t, q_t) = \mathbb{E}_{\mathbf{z}_t \sim q_t}[-\log P_\theta(\mathbf{x}_t|\mathbf{z}_t)] \approx \sum_{m=1}^{M} \mathcal{L}_{\text{state}}(\mathbf{x}_t, f_{\text{dec}}(\hat{\mathbf{z}}_t^{(m)})) \tag{1}$$

$$\mathcal{L}_{\text{pred}}(\mathbf{x}_{t+1}, \hat{q}_{t+1}) = \mathbb{E}_{\hat{\mathbf{z}}_{t+1} \sim \hat{q}_{t+1}}[-\log P_\theta(\mathbf{x}_{t+1}|\hat{\mathbf{z}}_{t+1})] \approx \sum_{m=1}^{M} \mathcal{L}_{\text{state}}(\mathbf{x}_{t+1}, f_{\text{dec}}(\hat{\mathbf{z}}_{t+1}^{(m)})) \tag{2}$$

$$\mathcal{L}_{\text{cons}}(\hat{q}_{t+1}, q_{t+1}) = \text{KL}(\hat{q}_{t+1}\|q_{t+1}) \tag{3}$$

$$\mathcal{L}_{\text{reg}}(q_t, q_{t+1}) = \text{KL}(q_t\|\mathcal{N}(\mathbf{0}, \mathbf{I})) + \text{KL}(q_{t+1}\|\mathcal{N}(\mathbf{0}, \mathbf{I})) \tag{4}$$

---

**Algorithm 1:** TactileVAD Training Algorithm

---

**Input:**
$K$: Number of samples
$T$: Number of timesteps
$(\mathbf{u}_0, \mathbf{u}_1, \ldots \mathbf{u}_{T-1})$: Initial control sequence

1   $f_{\text{lat\_dyn}} : \{\mathbf{A}^{(0)}, \mathbf{B}^{(0)}\} \leftarrow$ Initialization;

2   $f_{\text{dec}} : \{\theta_i^{(0)}\} \leftarrow$ Initialization;

3   $q : \{\boldsymbol{\mu}_i^{(0)}, \boldsymbol{\Sigma}_i^{(0)}\} \leftarrow$ Initialization;

4   $k \leftarrow 0$;

5   **while** *not converged* **do**

6     $[\mathbf{z}_t] \sim q(\mathbf{z}_t; \boldsymbol{\mu}_t^{(k)}, \boldsymbol{\Sigma}_t^{(k)})$ ;

7     $q(\hat{\mathbf{z}}_{t+1}; \hat{\boldsymbol{\mu}}_t^{(k)}, \hat{\boldsymbol{\Sigma}}_t^{(k)}) = \mathcal{N}(\mathbf{A}\boldsymbol{\mu}_t + \mathbf{B}\mathbf{u}_t, \mathbf{A}\boldsymbol{\Sigma}_t\mathbf{A}^\top)$;

8     $[\hat{\mathbf{z}}_{t+1}] \sim q(\hat{\mathbf{z}}_{t+1}; \hat{\boldsymbol{\mu}}_{t+1}^{(k)}, \hat{\boldsymbol{\Sigma}}_{t+1}^{(k)})$;

9     $[\mathbf{x}_t] = [f_{\text{dec}}(\mathbf{z}_{t+1})]$;

10    $[\hat{\mathbf{x}}_{t+1}] = [f_{\text{dec}}(\hat{\mathbf{z}}_{t+1})]$;

11    $l_{\text{rec}} = \mathbb{E}_{\mathbf{z}_t \sim q_t}[-\log P_\theta(\mathbf{x}_t|\mathbf{z}_t)]$;

12    $l_{\text{pred}} = \mathbb{E}_{\hat{\mathbf{z}}_{t+1} \sim \hat{q}_{t+1}}[-\log P_\theta(\mathbf{x}_{t+1}|\hat{\mathbf{z}}_{t+1})]$;

13    $l_{\text{cons}} = \text{KL}(q(\hat{\mathbf{z}}_{t+1}; \hat{\boldsymbol{\mu}}_t^{(k)}, \hat{\boldsymbol{\Sigma}}_t^{(k)})\|q(\mathbf{z}_t; \boldsymbol{\mu}_{t+1}^{(k)}, \boldsymbol{\Sigma}_{t+1}^{(k)}))$;

14    $l_{\text{kl}} = \text{KL}(q(\mathbf{z}_t; \boldsymbol{\mu}_{t+1}^{(k)}, \boldsymbol{\Sigma}_{t+1}^{(k)})\|P(Z))$;

15    $l = \alpha_{\text{rec}}l_{\text{rec}} + \alpha_{\text{pred}}l_{\text{pred}} + \alpha_{\text{cons}}l_{\text{cons}} + \alpha_{\text{kl}}l_{\text{kl}}$;

16    $\{\theta, \mathbf{A}, \mathbf{B}, \boldsymbol{\mu}_t, \boldsymbol{\Sigma}_t, \boldsymbol{\mu}_{t+1}, \boldsymbol{\Sigma}_{t+1}\}^{(k+1)} \leftarrow \underset{\theta, \mathbf{A}, \mathbf{B}, \boldsymbol{\mu}_t, \boldsymbol{\Sigma}_t, \boldsymbol{\mu}_{t+1}, \boldsymbol{\Sigma}_{t+1}}{\text{grad\_step}} (l)$;

17    $k \leftarrow k + 1$;

---

## A.2   TactileVAD Control Details

In this section, we expand on Section 4.4 to describe the integration details of the TactileVAD model with the controller. Given that we designed the latent space dynamics to be linear, we integrate TactileVAD's learned latent dynamics with the LQR control law directly. Algorithm 2 summarizes the control method employed. Note that the controller is formulated in the latent space $\mathcal{Z}$ and therefore needs to project the observations $\mathbf{x}$ to their respective latent values $\mathbf{z}$ to compute optimal actions (step 6 in Algorithm 2).

**Algorithm 2:** TactileVAD Control Algorithm

**Input:**
$\mathbf{x}_g$: Goal observation representing $\mathbf{s}_g$
$\mathbf{Q}$: quadratic state cost
$\mathbf{R}$: quadratic action cost
n: number of inference iterations

**1** $\mathbf{K} \leftarrow \text{LQRSolution}(\mathbf{Q}, \mathbf{R})$ ;
**2** $\mathbf{z}_g \leftarrow \text{inference}_{f_{\text{dec}}}(\mathbf{x}_g)$ ;
**3 while** *not converged* **do**
**4**     $\mathbf{x}_t \leftarrow \text{get\_observation}()$;
**5**     $\mathbf{z}_t \leftarrow \text{inference}_{f_{\text{dec}}}(\mathbf{x}_t)$ ;
**6**     $\mathbf{u}_t \leftarrow -\mathbf{K}(\mathbf{z}_t - \mathbf{z}_g)$;
**7**     $\text{send\_to\_actuator}(\mathbf{u}_t)$;
**8**     $k \leftarrow k + 1$;

# Appendix B    Experimental Details

## B.1    Baselines Details

All four benchmarked models share the same decoder architectures and latent dynamics parametrization. The encoder-decoder approaches (AE and E2C) share the same encoder structure. However, the key difference is in the latent elements. The non-variational approaches (AE and AD) latent elements take the form of vectors, i.e. $\mathbf{z} \in \mathbb{R}^{n_z}$, where the AE encoder computes the latent vector directly. The variational approaches (E2C and VAD) employ conditional distribution as latent representations. Therefore, the latent elements $\mathbf{z}$ are composed of the mean and covariance parameters of the conditional normal distribution: $\mathbf{z} = (\boldsymbol{\mu}, \text{diag}(\boldsymbol{\Sigma}))$ where $\mathbf{q}_t = \mathcal{N}(\boldsymbol{\mu}_t, \boldsymbol{\Sigma}_t)$. Therefore, the E2C encoder produces normal distribution parameters $\boldsymbol{\mu}$ and $\text{diag}(\boldsymbol{\Sigma})$, similar to a VAE. For the encoder-decoder approaches (AE and E2C) the latent elements are obtained via a forward call of the encoder. Decoder-only approaches (AD and VAD) optimize a codebook of latent vectors $\mathbf{z}$ and at run-time, they obtain the latent elements using inference via search as described in section 4.3. Fig. B.3 shows the schematic for AE over trajectories. Fig. B.2 shows the schematic for E2C over trajectories. Fig. B.3 shows the auto-decoder architecture.

The baselines, as well as our proposed model, are trained with a loss composed of four terms:

$$\mathcal{L} = \alpha_{\text{rec}}\mathcal{L}_{\text{rec}}(\mathbf{x}_t, \mathbf{z}_t) + \alpha_{\text{pred}}\mathcal{L}_{\text{pred}}(\mathbf{x}_{t+1}, \hat{\mathbf{z}}_{t+1}) + \alpha_{\text{cons}}\mathcal{L}_{\text{cons}}(\hat{\mathbf{z}}_{t+1}, \mathbf{z}_{t+1}) + \alpha_{\text{reg}}\mathcal{L}_{\text{reg}}(\mathbf{z}_t, \mathbf{z}_{t+1}) \tag{5}$$

For encoder-decoder approaches (AE and E2C) the latent elements are obtained via a forward call of the encoder. Decoder-only approaches (AD and VAD) optimize a codebook of latent vectors $\mathbf{z}$ and at run-time, they obtain the latent elements using inference via search as described in section 4.3.

Next, we will describe each loss term in equation 5:

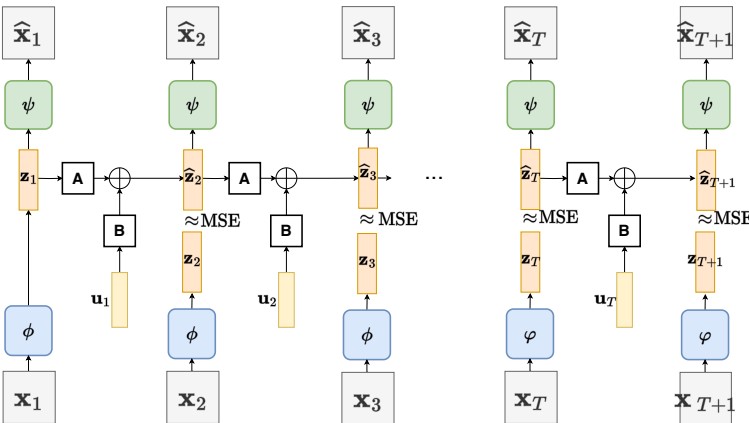

Figure B.1: **AE Model**

1. $\mathcal{L}_{\text{rec}}$ - **Reconstruction Loss**: Incentivizes the latent space to reconstruct the observed state $\mathbf{x}$, i.e. $\psi(\mathbf{z}_t) = \mathbf{x}_t$ and is formulated in the state observation space.

$$\mathcal{L}_{\text{rec}}(\mathbf{x}_t, \mathbf{z}_t) = \mathcal{L}_{\text{obs}}(\mathbf{x}_t, \psi(\mathbf{z}_t))$$

   - For encoder-decoder approaches (AE and E2C), the latent element is obtained via an encoder call. Therefore, this can also be expressed as:

$$\mathcal{L}_{\text{rec}}(\mathbf{x}_t, \mathbf{z}_t) = \mathcal{L}_{\text{obs}}(\mathbf{x}_t, \psi(\mathbf{z}_t)) = \mathcal{L}_{\text{obs}}(\mathbf{x}_t, \psi(\phi(\mathbf{x}_t)))$$

   - Decoder-only approaches (AD and VAD) optimize the codebook $\mathbf{z}_t$ via backpropagation.

2. $\mathcal{L}_{\text{pred}}$ - **Prediction Loss**: Incentivizes the next state prediction to reconstruct the expected next observed state. It is formulated in the state observation space.

$$\mathcal{L}_{\text{pred}}(\mathbf{x}_{t+1}, \hat{\mathbf{z}}_{t+1}) = \mathcal{L}_{\text{obs}}(\mathbf{x}_{t+1}, \psi(f_{\text{lat\_dyn}}(\mathbf{z}_t, \mathbf{u}_t)))$$

   - For encoder-decoder approaches (AE and E2C), the latent element is obtained via an encoder call. Therefore, this can also be expressed as:

$$\mathcal{L}_{\text{pred}}(\mathbf{x}_{t+1}, \hat{\mathbf{z}}_{t+1}) = \mathcal{L}_{\text{obs}}(\mathbf{x}_t, \psi(\mathbf{z}_t)) = \mathcal{L}_{\text{obs}}(\mathbf{x}_{t+1}, \psi(f_{\text{lat\_dyn}}(\phi(\mathbf{x}_t), \mathbf{u}_t)))$$

   - Decoder-only approaches (AD and VAD) optimize the codebook $\mathbf{z}_t$ via backpropagation.

3. $\mathcal{L}_{\text{cons}}$ - **Consistency Loss**: Incentivizes the latent space to be consistent with the imposed dynamics.

   - For variational approaches (E2C and VAD) this term takes the form of a KL divergence between the predicted latent distributions and the expected next latent distribution:

$$\mathcal{L}_{\text{cons}}(\hat{\mathbf{z}}_{t+1}, \mathbf{z}_{t+1}) = \text{KL}(\hat{\mathbf{z}}_{t+1} \| \mathbf{z}_{t+1})$$

   - For non-variational approaches (AE and AD), this term is the MSE between the expected next latent vector $\mathbf{z}_{t+1}$ and the estimated next latent vector $\hat{\mathbf{z}}_{t+1}$:

$$\mathcal{L}_{\text{cons}}(\hat{\mathbf{z}}_{t+1}, \mathbf{z}_{t+1}) = \text{MSE}(\hat{\mathbf{z}}_{t+1}, \mathbf{z}_{t+1})$$

4. $\mathcal{L}_{\text{reg}}$ - **Regularization Loss**: Regularizes the latent elements $\mathbf{z}$ so they are not arbitrary large.

   - For variational approaches (E2C and VAD) this term takes the form of a KL divergence between the latent distributions and the standard normal distribution $\mathcal{N}(\mathbf{0}, \mathbf{I})$:

$$\mathcal{L}_{\text{reg}}(\mathbf{z}_t, \mathbf{z}_{t+1}) = \text{KL}(\mathbf{z}_t \| \mathcal{N}(\mathbf{0}, \mathbf{I})) + \text{KL}(\mathbf{z}_{t+1} \| \mathcal{N}(\mathbf{0}, \mathbf{I}))$$

   - For non-variational approaches (AE and AD), this term is the L2 norm of the latent vectors:

$$\mathcal{L}_{\text{reg}}(\mathbf{z}_t, \mathbf{z}_{t+1}) = \|\mathbf{z}_t\|_2 + \|\mathbf{z}_{t+1}\|_2$$

### B.1.1  Encoder-Decoder vs Decoder-Only Inference

The AE and E2C baselines are instances of encoder-decoder architectures, while our approach (VAD) and the AD baseline are instances of decoder-only models. As seen in section 4, encoder-decoder models are sensitive to geometric aliasing, while decoder-only models are robust to this effect. In exchange for this ability, decoder-only models need to be embedded via optimization search as described in section 4.3, which is more computationally expensive than the forward encoder call required in encoder-decoder models. In this section, we quantify this difference in our setup for the Moving Block simulated task, described in Section 5.2. We measure the average time to obtain a batch of $B$ embeddings $\mathbf{z}_1, \ldots, \mathbf{z}_B$ corresponding to a series of observed states $\mathbf{x}_1, \ldots, \mathbf{x}_B$. Our setup is composed of a Nvidia RTX 2080 Ti GPU and an AMD Ryzen 9 3950X CPU. We report the average computation time to obtain a batch of $B \in [1, 10, 100, 1000, 10000]$ averaged over 100 samples. Table 4 summarizes the results.

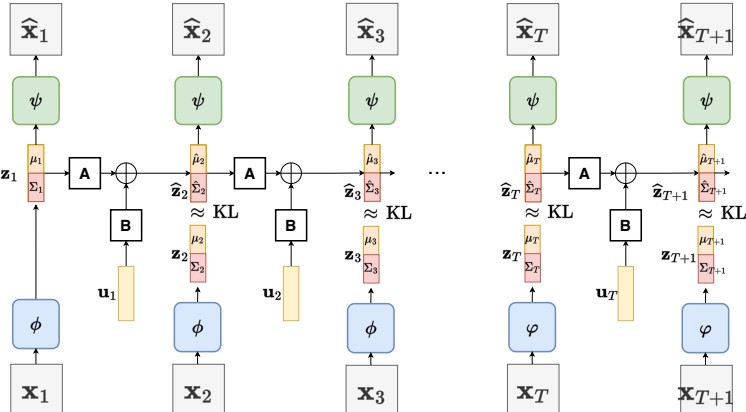

Figure B.2: **E2C Model**

| Method | Average Embedding Time [s] | | | | |
|---|---|---|---|---|---|
| | Batch Size | | | | |
| | 1 | 10 | 100 | 1000 | 10000 |
| Encoder-Decoder (AE) | $5.67 \cdot 10^{-5}$ | $5.66 \cdot 10^{-5}$ | $5.66 \cdot 10^{-5}$ | $5.68 \cdot 10^{-5}$ | $5.69 \cdot 10^{-5}$ |
| Decoder-only (AD) | 0.390 | 0.415 | 0.465 | 0.480 | 0.483 |

Table 4: **Encoder-decoder Embedding Computational Cost:** We compare the average time for obtaining an embedding between an encoder-decoder architecture (AE) and a decoder-only architecture (AD). The former encodes via a forward encoder call which is faster than the embedding-via-inference that the decoder-only architecture (AD) requires.

### B.1.2 Auto-Decoder (AD) vs Variational Auto-Decoder (VAD) Latent Space Structure

The auto-decoder (AD) baseline as well as our proposed method are instances of decoder-only architectures. By construction, both methods are robust against the geometric aliasing effects (see Sec. 4). However, the main difference between these approaches is the topology of the resulting embedding spaces. Prior work in image embedding and reconstruction [35, 37, 38] and learning latent space dynamics [26, 27] has shown how even in the presence of heavy regularization, the manifold of the embedding space of auto-encoder models is non-convex, takes on irregular geometry, contains holes, is often not smooth. F Fig. B.4 illustrates examples of these pathologies even for the simplest of dynamical systems. The key observation is the "jumbled" latent space of the AE model vs the much more well-structured and regular latent space of the variational approach. Paths within the latent space of the latter are better connected, regular, and inference within this space is significantly better posed. The consequence of AE pathologies is that trajectories in the latent space can easily become out of distribution and cumulative error leads to catastrophic failure. Further, learning latent space dynamics from the "jumble" and performing controls is significantly complicated. Beyond our setup, in the context of image compression, these failures correspond to poor and nonsensical reconstructions [35, 37, 38]. For dynamics, these issues result in severe divergences and instability [26, 27].

In contrast, variational approaches impose a much more explicit structure on the latent space, imposing a conditional normal distribution. This structure avoids many of the pathologies of vanilla auto-encoder approaches and has become one of the dominant approaches in encoding [35, 39]. In addition, the variational approach allows for principled sampling, out-of-distribution detection, and a more uniform distribution of latent states that is more amenable to dynamics propagation.

To evaluate the difference between an AD and a VAD embedding spaces, we perform inference on a set of known latent elements from the codebook and compare the inferred states to the ground truth ones. The inference is initialized with the known latent space embedding $\mathbf{z}_{\text{gth}}$ with additive Gaussian noise. To account for the difference in latent space scale, we define the noise level proportional to the latent space dispersion, obtained via the stored codebook. The latent space is given by $\mathbf{z}_{\text{gth}} \in \mathbb{R}^{n_z}$ for the AD and $\mathbf{z}_{\text{gth}} = (\boldsymbol{\mu}, \boldsymbol{\Sigma})$ for the VAD. We evaluate two metrics: i) the L2 distance between the inferred latent state $\mathbf{z}^*$ and the expected latent state $\mathbf{z}_{\text{gth}}$, and ii) the decoded observed state $\mathbf{x}^* = \psi(\mathbf{z}^*)$ similarity with the expected observed state $\mathbf{x}_{\text{gth}}$. For a fair comparison between the different latent spaces, we evaluate the similarity over the normalized latent elements so scales are comparable. Table 5 summarizes the results. The results show that the AD is more sensitive to the initial search values than the VAD suggesting that the latent space of the VAD is more smooth and well structured.

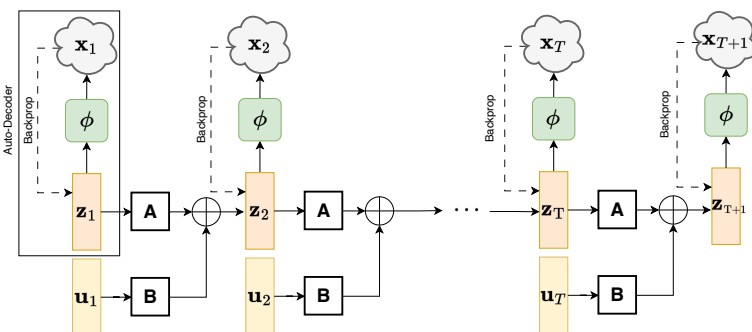

Figure B.3: **Auto-Decoder Latent Linear Dynamics Model (AD)**

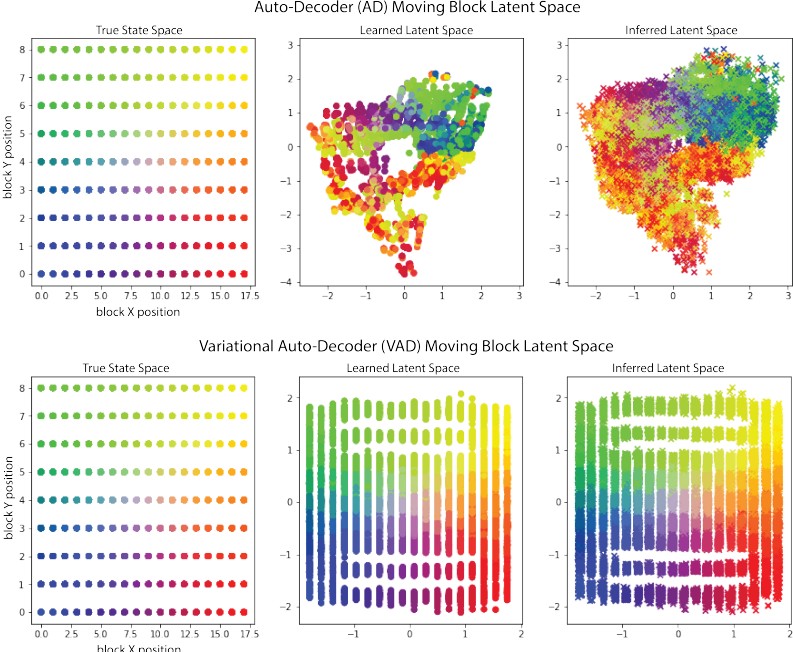

Figure B.4: **AD vs VAD Latent Space Inference**: (Top) AD learned latent space and inferred values for 0.1 noise level. (Bottom) VAD learned latent space and inferred values for 0.1 noise level. We plot the mean of the latent conditional distributions for the VAD. Color encodes the true state space location (left). We note pathological latent space of AD latent space which leads to poorer inference results. The key observation is how well-posed the latent space topology is (in terms of regularity, smoothness, and coverage) for VAD model. This more structured latent space results in better inference results since the inferred latent space values (right) match closer to the expected latent values (center).

| Noise Level | Latent Similarity Score ↓ | | State Similarity Score ↓ | |
|---|---|---|---|---|
| | AD | VAD | AD | VAD |
| 0.001 | $3.963 \cdot 10^{-4}$ | $2.108 \cdot 10^{-4}$ | $6.146 \cdot 10^{-4}$ | $1.485 \cdot 10^{-3}$ |
| 0.01 | $5.013 \cdot 10^{-4}$ | $2.358 \cdot 10^{-4}$ | $7.875 \cdot 10^{-4}$ | $1.487 \cdot 10^{-3}$ |
| 0.02 | $1.379 \cdot 10^{-3}$ | $3.0685 \cdot 10^{-4}$ | $1.220 \cdot 10^{-2}$ | $1.493 \cdot 10^{-3}$ |
| 0.05 | $9.765 \cdot 10^{-3}$ | $7.886 \cdot 10^{-4}$ | $0.3274$ | $1.534 \cdot 10^{-3}$ |
| 0.1 | $4.052 \cdot 10^{-2}$ | $2.601 \cdot 10^{-3}$ | $2.4107$ | $1.729 \cdot 10^{-3}$ |
| 0.2 | $0.1693$ | $0.01035$ | $7.317$ | $0.09373$ |
| 0.5 | $1.107$ | $0.0691$ | $16.35$ | $2.473$ |

Table 5: **VAD vs AD Latent Space Evaluation:** We evaluate the noise effect on the initial latent state estimation for the inferred latent similarity and the resultant decoded state.

## B.2 Moving Block Control Details

The moving block task is defined by $20 \times 20$ grid that contains a $3 \times 12$ block. The sensed area is $12 \times 8$ and is centered in the middle of the space. The true block state is defined by the block top corner coordinates $(x, y)$ The block motion is defined by limited on a box of size $u = (\delta x, \delta y) \in \mathcal{U} \subseteq [-3, 3] \times [-3, 3]$. States are encoded as binary masks The pose error is computed as the Manhattan distance. The models were trained on 500 random trajectories of length 20 steps. The reconstruction and prediction loss over observations is computed as the BCE against the ground truth states.

## B.3 Tactile Rod Grasping Details

The rod grasping task is defined over a $SE(2)$ space of robot-rod configurations. The space limits are $\mathcal{S} \subseteq [-20, 20] \times [-20, 50] \times [-\frac{\pi}{2}, \frac{\pi}{2}]$ (mm×mm×rad). Robot actions are constrained within a box defined as:

$$\mathbf{u} = (\delta x, \delta y, \delta \theta) \in \mathcal{U} \subseteq [-6, 6] \times [-5, 5] \times [-0.09\pi, 0.09\pi] \quad (\text{mm} \times \text{mm} \times \text{rad})$$

| Method | Bubbles Final Imprint Errors ($\cdot 10^{-6}$) [m] | | | | | | Gelslim Final Imprint Errors | | | | | |
|---|---|---|---|---|---|---|---|---|---|---|---|---|
| | 20mm Rod (Train) | | 15mm Rod | | 30mm Rod | | 20mm Rod (Train) | | 15mm Rod | | 30mm Rod | |
| | Mean ↓ | Std ↓ | Mean ↓ | Std ↓ | Mean↓ | Std↓ | Mean ↓ | Std ↓ | Mean ↓ | Std ↓ | Mean↓ | Std↓ |
| AE | 8.6062 | 4.3312 | 2.3818 | 2.1589 | 15.269 | 12.121 | 30.009 | 9.90 | 27.22 | 4.17 | 18.84 | 8.319 |
| E2C | 5.9678 | 5.2829 | 4.4046 | 4.6345 | 3.9464 | 2.0997 | 25.28 | 7.576 | 29.20 | 7.27 | 19.05 | 5.79 |
| AD | 6.5889 | 3.4163 | 3.3810 | 1.9211 | 11.281 | 3.8071 | 21.076 | 3.75 | 29.24 | 11.45 | 15.98 | 1.51 |
| VAD (ours) | **2.3723** | **1.0926** | **0.6295** | **0.2842** | **3.1057** | **2.67** | **19.631** | **12.252** | **23.91** | **9.44** | **14.52** | **6.34** |

Table 6: **Tactile Rod Grasping Evaluation: (Tactile similarity)**

For the bubbles sensors, states are encoded as deformation depth maps of size $(2, 25, 20)$. Gelslim data is encoded as color differences encoded as grayscale of size $(2, 20, 20)$

$$\text{Pose Score}(\begin{bmatrix} x_1 \\ y_1 \\ \theta_1 \end{bmatrix}, \begin{bmatrix} x_2 \\ y_2 \\ \theta_2 \end{bmatrix}) = \text{MSE}(x_1, x_2) + \text{MSE}(y_1, y_2) + r_g \text{MSE}(\theta_1, \theta_2) \quad (6)$$

where $r_g$ is the radius of gyration of the rod.

# Appendix C   Additional Experiment Results

## C.1   Moving Block Control

In this section, we provide further experiments for the moving block control simulation task. Figure C.1 shows instances of the reconstructed and predicted states for our method and the baselines. Note that encoder-based baselines suffer from aliasing.

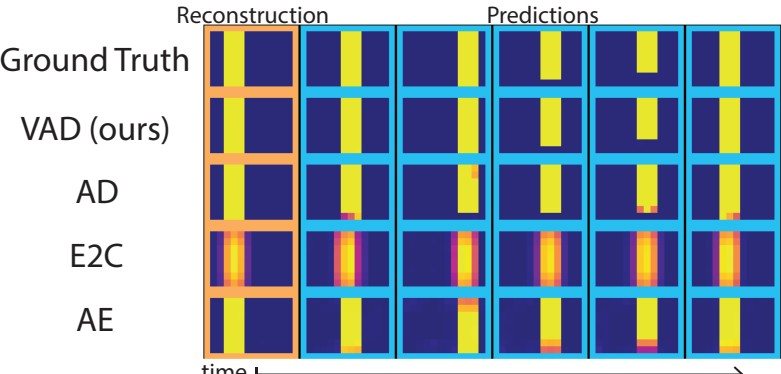

Figure C.1: **Moving Block Evaluation**: Reconstructed (orange) and predicted (blue) states from latent space inference and dynamic propagation. E2C and AE methods suffer from aliasing resulting on vertical uncertainty of the block position prediction.

## C.2   Tactile Rod Grasping

In this section, we provide further experiment results for the tactile rod grasping task. In particular, we evaluate how similar are the achieved states in terms of tactile observation similarity to the desired ones. Table 6 summarizes these results. To sum up, VAD produces better similarity results compared to baselines.

