# OpenReview forum: "TactileVAD: Geometric Aliasing-Aware Dynamics for High-Resolution Tactile Control"
_robot-learning.org/CoRL/2023/Conference — CoRL 2023 Poster_

### Official Review · Reviewer_rJSx · 2023-07-03

**Confidence:** 4
**Originality:** Good
**Technical Quality:** Good
**Clarity Of Presentation:** Good
**Impact:** 2

**Recommendation:**

Weak Accept: I recommend accepting the paper, but will not argue for my recommendation if the majority of other reviewers have a different opinion.

**Review:**

**Strengths**

- The method is described quite clearly and succinctly
- The paper’s original claim about better VAD performance in the presence of aliasing is well-supported by the moving block simulation experiment, and further validated by real-world experiments on tactile rod grasping and tactile cartpole.

**Weaknesses**

- Data diversity and quantity could be better evaluated. 500 trajectories seems like a large quantity of data.
- Control problems evaluated involve relatively easy control tasks, ie. static interaction or a 1-dimensional control task. Some analysis and insight into generalization to harder tasks would be useful
- Implementation details of baseline methods are missing. TactileVAD uses regularization terms like the L_KL loss term. Analysis on the criticality of this term as well as details of regularization for baseline methods will help improve confidence in the comparison numbers presented in the paper.
- Baseline methods seem significantly worse than the proposed method. Providing some intuition or explanation for this would be helpful. If not, adding more details, such as comparison of the individual loss terms would be useful and make the results more convincing. For instance, in table 2, using AD seems to do almost 10 times worse than VAD. Is this due to poor reconstruction, poor dynamics prediction, or is it just the model being underfit?
- Often, aliasing can be avoided by adding temporal information to the observation, for instance, by stacking frames. Comparisons to such approaches would provide some insight into whether aliasing is truly the problem being addressed by the use of VAD.
- The authors are requested to proofread the manuscript. There are a number of errors and typos, some of which are as follows:
    - None of the tables have units, which make the numbers difficult to understand
    - Line 246: Length of trajectories is “10”?
    - Figure 4 and Figure 6 have the same captions
    - There are also a number of other smaller typos

**Quality Of The Limitations Section:**

Limitations are addressed clearly

**Questions For Rebuttal:**

Please address the weaknesses described in the main review. In general, shedding some light on why and where the baseline methods fail, providing more details on the training of baseline methods, and adding units to the evaluation metrics would be quite useful in more convincingly demonstrating that TactileVAD is better than baseline methods.

**Robotics Focus:**

Sufficient demonstration on hardware

**Summary Of Paper:**

This paper uses a Variational Auto-Decoder model to learn linear latent dynamics models using tactile observation data from vision-based tactile sensors. The paper claims that unlike existing control approaches that involve learning a latent dynamics model, VADs overcome the problem of tactile geometric aliasing, ie. when different configurations or contact geometries result in the same tactile signal. The learned latent linear dynamics model is then used with an LQR controller on 1 simulated and 2 real-world tasks and compared with other alternatives for learning latent dynamics models.

**Summary Of Recommendation:**

While the idea in the paper seems interesting and they demonstrate their method in real robot experimental settings, comparative evaluations are poorly presented and lacking details making it difficult to conclusively believe the extent to which the presented method improves on existing methods.

---

### Official Review · Reviewer_bqGm · 2023-07-07

**Confidence:** 4
**Originality:** Very Good
**Technical Quality:** Good
**Clarity Of Presentation:** Good
**Impact:** 4

**Recommendation:**

Weak Accept: I recommend accepting the paper, but will not argue for my recommendation if the majority of other reviewers have a different opinion.

**Review:**

Strengths:
- The problem is well-motivated – geometric aliasing is an important problem in tactile dexterous manipulation, and the proposed method is a promising approach to addressing this aliasing problem.
- The experimental settings are very well designed: the simulated toy example is excellent for illustrating the problem without introducing any confounding factors, and multiple real-world experimental setups make the results more convincing.

Weaknesses:
- While an important advantage of the authors’ method is to be able to generate multiple candidate latent vectors at inference time (which was the main weakness of encoder-based methods in the first place), it’s not discussed how they actually perform this, or how it is integrated into the control system. Without this, it’s not intuitive that the method will actually improve over methods like the VAE encoder that suffer from the aliasing issue.
- I’m confused about the results of the auto-decoder dynamics baseline which seems like it should be much more competitive with the full tactileVAD method. Please see questions for additional details.
- As the authors themselves mention, this approach causes inference of the latent vectors to be significantly more computationally expensive than using a learned encoder. While I appreciate the authors’ transparency about this limitation, I would appreciate it if they quantified this difference so that readers could fully understand the properties of tactileVAD.


**Quality Of The Limitations Section:**

Additional details required

**Questions For Rebuttal:**

- I’m a bit confused about the auto-decoder dynamics baseline – my understanding is that the tactileVAD codebook optimizes the parameters of a normal distribution for each timestep in the training dataset. If this is the case, what does the AD baseline do? Does it optimize a single z vector for each timestep in the training dataset? If so, why is the performance of AD so much worse compared to the full method? It seems that such an approach should also inherit the anti-aliasing properties of the full method, as these representations would be optimized such that they can predict dynamics accurately. Additionally, in Section 4.1 it’s stated that “ Additionally, we choose to model the latent space as conditional distributions, which leads to more structured and smooth learned latent spaces.” but this statement does not appear to be justified by any experimental results.
- In section 4.3 “Embedding via Inference Search”, it’s stated that the inference approach offers the ability “to generate multiple candidate latent states that result in the same observation”, but it’s not discussed how this is performed – is gradient descent performed from varying initializations? Once multiple candidates are found, how are they used in the LQR controller? This is also not discussed in section 4.4.


Nitpicks:
- The variable $N$ is used in Section 4.1 (L137) but not defined prior.
- Figure 3 is slightly confusing – I am assuming that the left figure is showing the “full” training picture since it is being trained on multiple timesteps at the same time, while the right side shows the additional losses but on a single timestep for less clutter. But it’s not that clear that this is the case as the caption for (left) just says “trajectory generation of length T” when it seems like it is training (as backprop happens to infer $\mu$ and $\Sigma$ at every timestep)


**Robotics Focus:**

Sufficient demonstration on hardware

**Summary Of Paper:**

This paper proposes TactileVAD, a dynamics model that takes a decoder-only approach to reduce geometric aliasing. This is particularly important for tactile manipulation applications where aliasing is a significant challenge as tactile sensors can usually only make contact with a fraction of an object. When combined with learned linear dynamics and LQR control, the method is tested on a simple simulated setting of moving block control, a real-world tactile rod grasping task, and a real-world tactile cartpole task where the robot holds onto a tool handle attached to an actuated cart, and tries to keep the rod vertical. Experiments show that this approach outperforms auto-encoder and E2C based dynamics.

**Summary Of Recommendation:**

I think this work is well-motivated and has convincing experimental setups. However, it’s still not entirely clear to me how the particular way the VAD model is used is addressing the aliasing problem, and the auto-decoder baseline/ablation is confusing. Thus I don’t think this version of the manuscript should be accepted, but I would be happy to raise my score to accept, if these concerns are addressed and the paper revised.

**Update after rebuttal**: The authors have addressed my main concerns by expanding upon critical details about the method and by providing additional understanding for the performance of the baselines in the new Appendix B.1.1, B.1.2, and Figure B.4. Thus I am inclined to increase my score, as while I think that the experiments could still be more thorough, the idea of using an auto-decoder only model for a tactile control application is quite interesting.

---

### Official Review · Reviewer_pV8K · 2023-07-19

**Confidence:** 5
**Originality:** Good
**Technical Quality:** Very Good
**Clarity Of Presentation:** Very Good
**Impact:** 3

**Recommendation:**

Weak Accept: I recommend accepting the paper, but will not argue for my recommendation if the majority of other reviewers have a different opinion.

**Review:**

Strengths:
- The paper is well presented and makes its points clealy.
- Authors use decoder-only model giving better latent spaces compared with encoder-decoder model.
- Two different sensing modalities, Bubbles ( point cloud) and Gelslim (RGB), are both evaluated with the proposed method, and the results are acceptable.

Weaknesses:
- There is a lack of variation in the selection of objects. Authors only provides cylindrical object for grasping and do not try other objects such as prisms, cones, spheres or irregular shapes.


**Quality Of The Limitations Section:**

Limitations are addressed clearly

**Questions For Rebuttal:**

1. There is a lack of variation in the selection of objects. Authors only provides cylindrical object for grasping and do not try other objects such as prisms, cones, spheres or irregular shapes. Is it possible to increase the types of objects?

**Robotics Focus:**

Sufficient demonstration on hardware

**Summary Of Paper:**

This paper provides a tactile representation and controller to disambiguate geometric aliasing, and demonstrate it on two different sensing modalities. A benchmarking setup is also proposed to evaluate the stability of the control during perturbations.

**Summary Of Recommendation:**

Overall the paper is clear and gives a acceptable method to imporve the geometric reconstruction and prediction during grasping. Accept.

---

### Decision · Program_Chairs · 2023-08-30

**Decision:**

Accept (Poster)

**Comment:**

The paper presents a method that alleviates the issue of geometric aliasing in tactile manipulation. The idea is to use the decoder only. Because the latent space can be learned so that there is no aliasing, it is possible to learn the decorder even when there is tactile aliasing issues.

The reviewers raised a common concern. First, the experiment only contains a cylinderical object. Even though the authors argued that it is sufficient to demonstrate their contribution, I do not believe so because there are different challenges associated with the object shape. A diverse set of symmetrical objects will show the limitations of this method and what has to studied further.

I believe that the video should be significantly improved. The video should be self-explanatory. Now I do not see what the experiment is about.

The strength of this paper is that the main idea is simple but has significant impacts to grasping tasks. The authors rebuttal makes it clear about this point.

Even though this paper has a clear weakness, no reviewers are against it and I am willing to accept this paper.